

# Association between serum lipid and papillary thyroid cancer: a retrospective study in China

Zike Zhang[1,*], Xingyu Lan[2,*], Long You[1], Dongsheng Han[1], Hui Tang[3], Ying Zhao[1] and Xiao Hu[1]

[1] Department of Laboratory Medicine, The First Affiliated Hospital, Zhejiang University School of Medicine, Hangzhou, Zhejiang, China
[2] The First Division Hospital of Xinjiang Production and Construction Group in China, Akesu, Xinjiang, China
[3] Department of Pathology, The First Affiliated Hospital, Zhejiang University School of Medicine, Hangzhou, China
* These authors contributed equally to this work.

Corresponding author
Xiao Hu, huxiao700@zju.edu.cn

## ABSTRACT

The incidence of papillary thyroid cancer (PTC) has increased drastically in recent decades. Various studies have reported a concurrent rise in PTC morbidity in the obese, implying a possible role of lipids in the pathogenesis of PTC. However, the role of serum lipids in the pathogenesis of PTC requires further investigation. This study aimed to investigate the correlation between serum lipid levels and PTC. From January 1, 2019, to December 31, 2020, 1,650 PTC and 882 control samples were enrolled for this study. PTC subjects were more likely to have higher body mass index (BMI), fasting blood glucose (FBG) levels, triglyceride (TG) levels, and decreased high-density lipoprotein cholesterol (HDL-C) when compared to controls ($P < 0.05$). Although age and low-density lipoprotein cholesterol (LDL-C) did not appear to change across all age groups, the PTC patients exhibited pronounced differences in terms of TG and HDL-C when compared to controls ($P < 0.05$) for each age group. BMI (odds ratio, OR and 95% CI 1.045 [1.002–1.089], $P < 0.038$), FGB (OR and 95% CI 2.543 [1.968–3.286], $P < 0.001$), TG (OR and 95% CI 1.267 [1.025–1.566], $P < 0.001$), and HDL-C (OR and 95% CI 0.422 [0.289–0.616], $P < 0.001$) were risk factors of PTC in the multivariate analysis of females. For males, FBG (OR and 95% CI 2.136 [1.551–2.941], $P < 0.001$), TG (OR and 95% CI 1.264 [1.039–1.615], $P < 0.05$), total cholesterol (TCH) (OR and 95% CI 0.778 [0.626–0.968], $P < 0.001$), and HDL-C (OR and 95% CI 0.154 [0.077–0.308], $P < 0.001$) were risk factors of PTC. Both in the female and male subgroups, patients with tumors > 1 cm in size and multifocality had a greater risk of lymph node metastasis (LNM) among PTC subjects ($P < 0.001$). The study results revealed that elevated TG and declined HDL-C were related to increased PTC risk among Chinese of both sexes.

## INTRODUCTION

One of the most prevalent endocrine tumors is thyroid cancer (TC), which has various histological types, including papillary thyroid cancer (PTC, making up approximately 85% of TC (*Fagin & Wells, 2016*)), follicular thyroid cancer (FTC), anaplastic thyroid cancer (ATC), and medullary thyroid cancer (MTC). Over the past few decades, TC incidence has dramatically increased across the globe. In the United States, according to estimates, the total incidence of TC grew by 3% each year between 1974 and 2013 (*Qian et al., 2019*). Likewise, the age-standardized incidence of TC grew by an average of 3% per year from 3.21/100,000 in 2005 to 9.61/100,000 in 2015 in China (*Wang et al., 2020*). The increase in operations connected to treatment and follow-up is directly proportional to the rise in TC incidence, which has resulted in a fast-rising disease burden (*Deng et al., 2020*; *Janovsky et al., 2018*). Therefore, exploring the underlying pathogenesis of TC is in great need.

More comprehensive thyroid examinations, such as ultrasonography, mainly involve observing the incidence (*Lim et al., 2017*). Several risk factors for thyroid cancers have been hypothesized, including genetics, radiation, iodine, autoimmune illnesses, and environmental variables (*Tang et al., 2020*). Meanwhile, numerous studies have noted a concurrent rise in TC morbidity in obesity (*Pappa & Alevizaki, 2014*). *Xu et al. (2014)* found that in a group of Americans, Italians, and Germans, body fat percentage and body mass index (BMI) were strongly linked to a higher risk of PTC. Obesity and papillary, follicular, and undifferentiated TCs hold a significant positive association, according to a meta-analysis of 12,199 TC cases (*Schmid et al., 2015*). Serum lipids are strongly associated with obesity together with dyslipidemia. Abnormalities in a group of lipoproteins that include high levels of triglycerides (TG), total cholesterol (TCH), low-density lipoprotein cholesterol (LDL-C), or low levels of high-density lipoprotein cholesterol (HDL-C) are common features of several cancers, including breast and clear cell kidney cancer (*Yang et al., 2015*; *Ho et al., 2021*). However, the association between serum lipids and PTC is still incompletely understood, especially in the Chinese population.

This study used population-based samples from China to investigate the correlation between serum lipids and the extent of PTC at diagnosis or pathological characteristics like tumor size, multiplicity, and lymph node metastasis (LNM) of PTC. The study could contribute to further investigation into diagnosing and preventing papillary thyroid cancer, the most common type of thyroid cancer.

## MATERIALS AND METHODS

Portions of this text were previously published as part of a preprint https://www.researchsquare.com/article/rs-3444881/v1.

### Study design and patients

This study retrospectively analyzed patients with newly diagnosed PTC who had radical surgical resection at the First Affiliated Hospital, Zhejiang University School of Medicine, between January 1, 2019, and December 31, 2020. Patients who fulfilled the following criteria for inclusion were recruited: (1) age $\geqq$ 18 years; (2) primary PTC was

pathologically confirmed; (3) without hypolipidemic agent treatment history; (4) without hepatic, renal insufficiency, diabetes, or other kinds of disorders; (5) with complete clinical and pathological information. The control group comprised individuals with benign nodules (Thyroid Imaging Reporting and Data System (TI-RADS) scores ≤ 2) of comparable ages and sexes who completed routine medical examinations throughout the same period. These individuals had no additional medical history that would have affected their serum lipid levels or thyroid function.

Demographic, clinical, laboratory, surgical, and pathological data of enrolled patients were gathered from the database created by the First Affiliated Hospital, School of Medicine, Zhejiang University. The primary clinical data included serum lipid levels (TG, TCH, HDL-C, LDL-C), fasting blood glucose (FBG) levels, serum thyroid stimulating hormone (TSH), total triiodothyronine (TT3), total thyroid hormone (TT4), free triiodothyronine (FT3), and free thyroid hormone (FT4). Information about LNM was gathered from records of surgery and pathology. The counts of tumors were counted by pathological and ultrasonography reports, in which patients with a single tumor and two or more tumors were considered unifocal and multifocal, respectively.

After a 12-h fast, all venous blood samples were collected in the morning. The immunoluminescent technique (Abbott i2000, Wiesbaden, Germany) was used for the thyroid function test. TG, TCH, HDL-c, LDL-c, and FBG were examined by a Cobas 8000 Clinical Analyzer (Cobas, New York, NY, USA) using assay-specific Roche reagents (Roche Diagnostics, Indianapolis, IN, USA). Two different researchers independently collected the data. We accessed the date between January 1, 2019 and December 31, 2020. Anonymization was conducted prior to further analysis.

## Statistical analysis

Continuous normally and non-normally distributed data were given as mean ± standard deviation (SD) and median (25th–75th percentile). Percentages were used to represent categorical variables. The frequencies for categorical data were compared by the *chi*-square test, whereas the continuous variables were compared using one-way ANOVA, Mann-Whitney test, and Student's *t*-test by IBM SPSS (version 26.0, Armonk, NY, USA). The effects of blood lipid levels on clinical features were assessed using logistic regression univariate and multivariate analysis, which computed odds ratios and their accompanying 95% confidence intervals (CIs). *P*-values were two-sided, and statistical significance was given to values below 0.05.

## Ethics approval and consent to participate

This work was approved by the Ethics Committee of the First Affiliated Hospital of Medical College at Zhejiang University, the reference number IIT20230089A. This was a retrospective study, and the results were anonymized; therefore, informed consent was not required to use the samples and data.

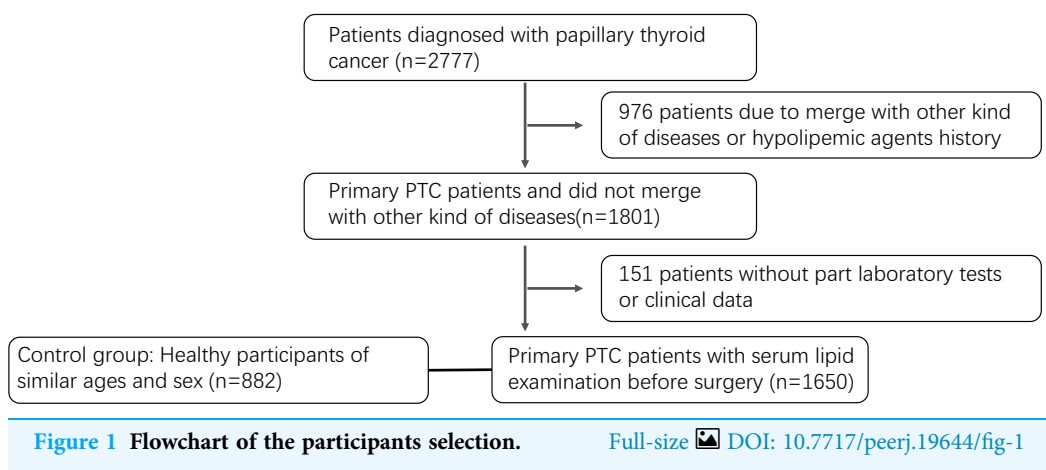

**Figure 1 Flowchart of the participants selection.**

**Table 1 Clinical variables of PTC patients.**

| Variables | Female | | | | | Male | | | | | P value |
|---|---|---|---|---|---|---|---|---|---|---|---|
| | Total (*n* = 1,214) | 0–34 (*n* = 228) | 35–55 (*n* = 696) | 56– (*n* = 290) | P value | Total (*n* = 436) | 0–34 (*n* = 116) | 35–55 (*n* = 232) | 56– (*n* = 88) | P value | |
| **Symptoms (*n*)**[a] | 98 (8.07) | 9 (3.95) | 59 (8.48) | 30 (10.34) | 0.026 | 28 (6.42) | 6 (5.17) | 14 (6.03) | 8 (9.09) | 0.496 | 0.157 |
| **Course (months)** | 1.24 | 0.97 | 1.15 | 1.71 | 0.073 | 1.1 | 0.82 | 1.1 | 1.47 | 0.278 | 0.258 |
| **Tumour size (%)** | | | | | | | | | | | |
| ≤1 cm | 936 | 163 | 553 | 220 | | 320 | 78 | 179 | 63 | | |
| >1cm | 278 (22.9) | 65 (28.51) | 143 (20.55) | 70 (24.14) | 0.039 | 116 (26.61) | 38 (32.76) | 53 (22.84) | 25 (28.41) | 0.13 | 0.119 |
| **Nodule number (%)** | | | | | | | | | | | |
| Unifocal (*n* = 1) | 724 | 159 | 412 | 153 | | 291 | 83 | 157 | 51 | | |
| Multifocal (*n* > 1) | 490 (40.36) | 69 (30.26) | 284 (40.8) | 137 (47.24) | <0.001 | 145 (33.26) | 33 (28.45) | 75 (32.33) | 37 (42.05) | 0.113 | 0.009 |
| **LNM (%)** | | | | | | | | | | | |
| No | 786 | 109 | 462 | 215 | | 213 | 39 | 124 | 50 | | |
| Yes | 428 (35.26) | 119 (52.19) | 234 (33.62) | 75 (25.86) | <0.001 | 223 (51.15) | 77 (66.38) | 108 (46.55) | 38 (43.18) | 0.001 | <0.001 |
| **LNR (%)** | 35.8 | 37.95 | 34.97 | 34.99 | 0.21 | 48.48 | 53.92 | 46.94 | 41.9 | 0.041 | <0.001 |

Notes:
LNM, Lymph Node Metastasis; LNR, Lymph node ratio, proportion of positive lymph nodes *vs* sampled lymph nodes.
[a] The number of patients with hoarseness, sore throat, dysphagia, fever, or other symptoms.

# RESULTS

## Clinical characteristics of enrolled participants

This study includes 882 control samples and 1,650 PTC patients (Fig. 1). Table 1 shows the baseline features of the PTC subjects. The average age of PTC subjects and control group were 45.5 ± 11.8 and 45.1 ± 11.5 (PTC females: 46.1 ± 11.7, PTC males: 43.9 ± 12.1; control females: 45.5 ± 11.4, control males: 44.5 ± 11.4). The age groups with the highest percentage of patients were those between the ages of 35 and 55 (56.24%), followed by those above the age of 55 (22.91%) and those under the age of 34 (20.85%). In comparison to the control group, which had a BMI of 22.72 ± 2.92 (21.76 ± 2.71 for females and

24.01 ± 2.83 for males), the BMI of PTC patients was 23.27 ± 3.24 (22.73 ± 3.08 for females and 24.77 ± 3.20 for male). The mean disease course was 1.2 months, with 1.24 months in women and 1.1 months in males (*P* = 0.258). Hoarseness, sore throat, dysphagia, fever, or other symptoms affected 125 (7.58%) patients (female: 8.07%; male: 6.42%, *P* = 0.157). Age increased the incidence of complications in women (*P* < 0.05). With a median tumor size of 0.87 cm and a 0.1–6 cm range, 1256 instances (76.12%) were 1 cm or smaller. The proportion of patients with larger tumors (>1 cm) was elevated in younger female patients (*P* < 0.05); a similar tendency was seen in men without any apparent difference (*P* = 0.13). Overall, 1,015 patients (61.52%) had only one tumor, and 635 (38.48%) had two or more tumors. Older female patients had a higher percentage of multifocal instances (*P* < 0.001), while men showed a similar trend but with no statistically significant difference (*P* = 0.113). A total of 651 patients (39.45%) had LNM, which was confirmed by pathological findings, among which men and younger patients had higher percentages (*P* < 0.001). Female patients had a mean lymph node ratio (LNR, ratio of positive lymph nodes *vs.* sampled lymph nodes) of 35.8% *vs.* 48.48% (*P* < 0.001), while younger patients had a greater LNR.

### Serum lipid levels exhibited significant differences in PTC patients

Higher BMI, FBG levels, TG levels, and poorer HDL-C were also associated with patients with PTC compared to control participants (*P* < 0.05) (Fig. 2). Although age and LDL-C were unchanged across all age groups, individuals with PTC exhibited obvious differences in TG and HDL-C compared to controls (*P* < 0.05) for each age group. BMI was considerably greater in PTC than in healthy individuals in both the youth and middle-aged categories (*P* < 0.05); however, no apparent difference was observed in the elder group. TCH was considerably lower in the young female PTC patients (*P* < 0.05) (Table 2). Overall, PTC patients had dysregulated serum lipid levels compared to the control group, suggesting a possible role of serum lipids in the pathogenesis of PTC.

### Univariable and multivariate analysis showed significant associations between serum lipid levels and PTC incidence

Univariate and multivariate logistic regression models were carried out to further explore the relationship between each serum lipid indicator and PTC. An increased odds ratio (OR) of PTC was linked to BMI (OR and 95% CI 1.117 [1.075–1.160], *P* < 0.001), FBG (OR and 95% CI 2.57 [2.025–3.262], *P* < 0.001), and TG (OR and 95% CI 1.801 [1.490–2.176], *P* < 0.001) in the univariate analysis for females, whereas a decreased OR of PTC was linked to HDL-C (OR and 95% CI 0.305 [0.223–0.418], *P* < 0.001). For males, BMI (OR and 95% CI 1.09 [1.038–1.144], *P* = 0.001), FBG (OR and 95% CI 2.059 [1.540–2.754], *P* < 0.001), and TG (OR and 95% CI 1.689 [1.392–2.048], *P* < 0.001) were positively associated with the risk for PTC, while TCH (OR and 95% CI 0.786 [0.660–0.935], *P* < 0.001), HDL-C (OR and 95% CI 0.096 [0.055–0.169], *P* < 0.001) were in negative correlations. The significant variables (*P* < 0.05) were selected for the multivariate analysis. BMI (OR and 95% CI 1.045 [1.002–1.089], *P* < 0.038), FBG (OR and 95% CI 2.543

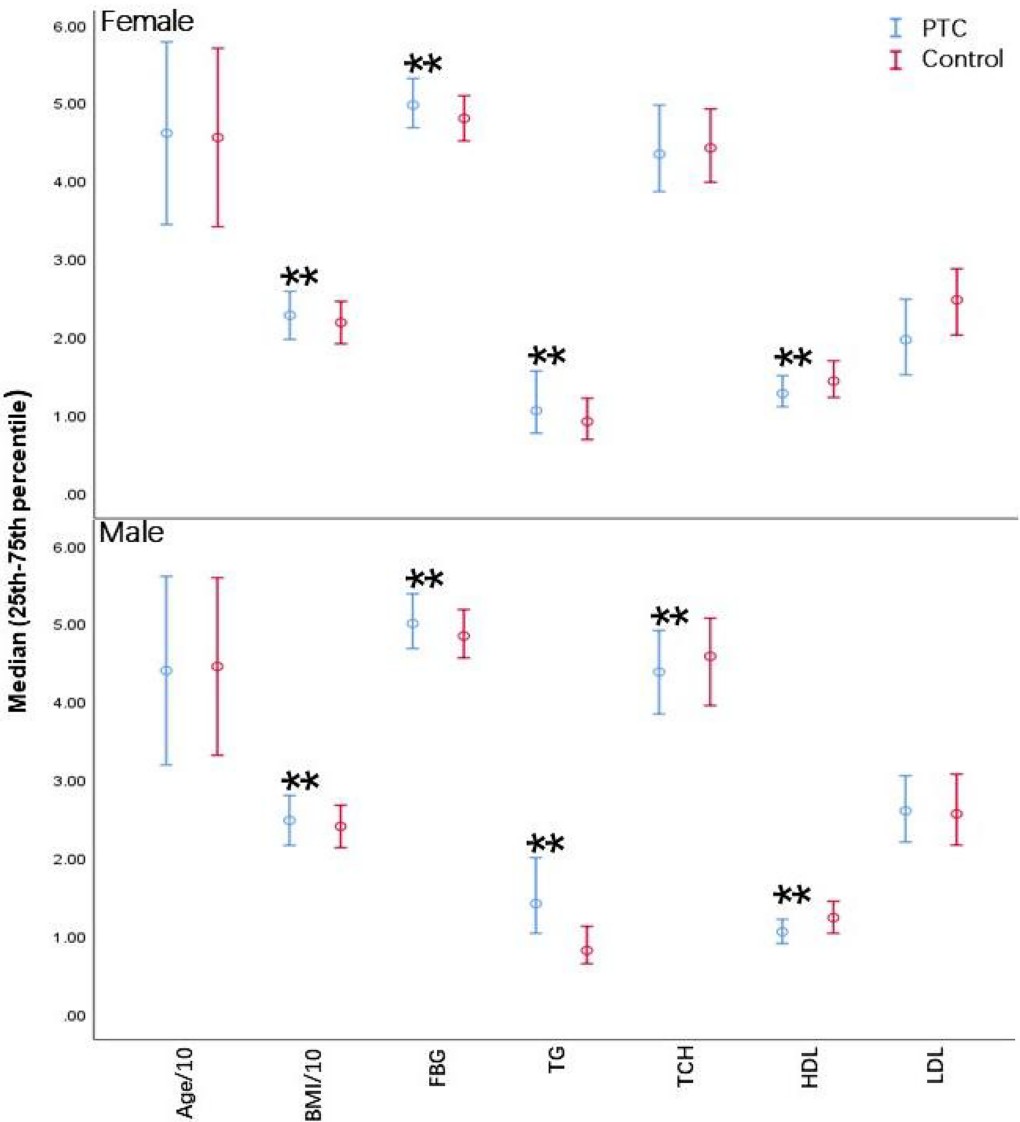

**Figure 2 Differences in biochemical parameters between PTC patients and controls.** BMI, Body Mass Index; FBG, fasting blood glucose; TG, triglycerides; TCH, total cholesterol; HDL, high-density lipoprotein cholesterol; LDL, low-density lipoprotein cholesterol. Error bars represent the interquartile range (25th–75th percentile) for non-normally distributed data. $^{**}P < 0.001$.

[1.968–3.286], $P < 0.001$), TG (OR and 95% CI 1.267 [1.025–1.566], $P < 0.001$), and HDL-C (OR and 95% CI 0.422 [0.289–0.616], $P < 0.001$) remained as risk factors of PTC in the multivariate analysis of females. Regarding males, FBG (OR and 95% CI 2.136 [1.551–2.941], $P < 0.001$), TG (OR and 95% CI 1.264 [1.039–1.615], $P < 0.05$), TCH (OR and 95% CI 0.778 [0.626–0.968], $P < 0.001$), and HDL-C (OR and 95% CI 0.154 [0.077–0.308], $P < 0.001$) were still risk factors of PTC. These results are presented in Table 3.

**Table 2 Baseline variables of the study population by age and gender groups.**

| Variables | Youth group (18–34 years) | | | | Middle-aged group (35–55 years) | | | | Middle-aged and elderly group (≥56 years) | | | |
|---|---|---|---|---|---|---|---|---|---|---|---|---|
| | Female | | Male | | Female | | Male | | Female | | Male | |
| | Cases (n = 228) | Controls (n = 100) | Cases (n = 116) | Controls (n = 86) | Cases (n = 696) | Controls (n = 319) | Cases (n = 232) | Controls (n = 191) | Cases (n = 290) | Controls (n = 109) | Cases (n = 88) | Controls (n = 77) |
| Age (year) | 29.6 ± 3.5 | 29.9 ± 2.5 | 29.8 ± 3.0 | 30.3 ± 2.6 | 45.5 (39.3–51) | 44 (40–50) | 44.4 ± 6.5 | 44.5 ± 5.9 | 61.3 ± 5.1 | 62.2 ± 4.6 | 61.6 ± 5.1 | 60.5 ± 4.4 |
| BMI | 21.99 ± 3.19 | 20.37 ± 2.45** | 24.6 ± 3.65 | 23.34 ± 2.63* | 22.73 ± 3.02 | 21.87 ± 2.54** | 25.03 ± 3.03 | 24.26 ± 2.76* | 23.30 ± 3.01 | 22.94 ± 2.86 | 24.27 ± 2.97 | 24.10 ± 2.63 |
| FBG (mmol/L) | 4.80 (4.55–5.08) | 4.68 (4.44–4.97)* | 4.78 (4.52–5.09) | 4.78 (4.58–5.07) | 4.98 (4.71–5.31) | 4.81 (4.51–5.06)** | 5.04 (4.73–5.40) | 4.82 (4.56–5.20)** | 5.10 (4.78–5.45) | 4.91 (4.56–5.24)** | 5.27 (4.95–5.67) | 4.92 (4.54–5.22)** |
| TG (mmol/L) | 0.81 (0.64–1.12) | 0.72 (0.59–0.94)* | 1.35 (1.04–1.95) | 0.90 (0.71–1.34)** | 1.25 ± 0.80 | 1.04 ± 0.60** | 1.47 (1.03–2.12) | 1.22 (0.79–1.75)** | 1.39 (1.01–1.86) | 1.09 (0.81–1.45)** | 1.39 (0.99–1.82) | 1.13 (0.86–1.57)* |
| TCH (mmol/L) | 4.01 (3.63–4.56) | 4.19 (3.74–4.71)* | 4.21 (3.64–4.67) | 4.35 (3.88–4.84) | 4.39 ± 0.82 | 4.44 ± 0.72 | 4.45 (3.93–4.97) | 4.58 (3.95–5.12) | 4.84 (4.26–5.44) | 4.70 (4.27–5.35) | 4.63 (3.84–5.13) | 4.72 (4.08–5.24) |
| HDL-C (mmol/L) | 1.34 (1.14–1.51) | 1.52 (1.29–1.80)** | 1.02 (0.87–1.17) | 1.26 (1.09–1.50)** | 1.27 (1.09–1.52) | 1.41 (1.21–1.66)** | 1.03 (0.9–1.2) | 1.19 (1.00–1.39)** | 1.26 (1.08–1.46) | 1.40 (1.19–1.66)* | 1.12 (0.95–1.30) | 1.26 (1.07–1.45)* |
| LDL-C (mmol/L) | 2.2 (1.93–2.71) | 2.25 (1.88–2.67) | 2.53 (2.05–2.80) | 2.42 (2.17–2.78) | 2.52 ± 0.68 | 2.46 ± 0.58 | 2.64 (2.22–3.1) | 2.62 (2.14–3.10) | 2.86 (2.39–3.33) | 2.76 (2.28–3.28) | 2.70 (2.28–3.17) | 2.68 (2.18–3.11) |
| TSH (mIU/L) | 1.49 (1.10–2.12) | 1.48 (1.04–2.07) | 1.23 (0.89–1.78) | 1.66 (1.25–2.18)** | 1.41 (1.00–2.05) | 1.70 (1.18–2.23)** | 1.27 (0.96–1.71) | 1.49 (1.12–2.01)* | 1.42 (1.01–1.99) | 1.76 (1.24–2.63)** | 1.32 (0.67–1.95) | 1.51 (1.00–2.00) |
| TT4 (nmol/L) | 101.18 (90.10–111.41) | 96.76 (89.37–103.61)* | 102.12 (90.78–112.29) | 97.99 (85.69–107.37)* | 100.40 (90.72–109.74) | 94.79 (85.93–104.07)** | 98.91 (90.06–111.69) | 96.60 (85.44–105.40)* | 98.22 (88.28–110.76) | 101.26 (89.85–116.05) | 99.69 (87.78–111.69) | 90.82 (81.32–102.80)* |
| TT3 (nmol/L) | 1.54 (1.39–1.69) | 1.52 (1.38–1.63) | 1.50 (1.38–1.62) | 1.62 (1.52–1.77)** | 1.53 (1.39–1.67) | 1.42 (1.30–1.56)** | 1.52 (1.38–1.66) | 1.62 (1.46–1.77)** | 1.55 (1.41–1.70) | 1.54 (1.43–1.68) | 1.51 (1.31–1.66) | 1.58 (1.45–1.79)* |
| FT4 (pmol/L) | 12.97 (12.10–13.85) | 13.16 (12.30–13.87) | 13.39 (12.57–14.44) | 13.54 (12.63–14.63) | 12.91 (12.11–13.78) | 12.67 (11.85–13.36)* | 13.18 (12.16–14.18) | 12.85 (12.24–13.59)* | 12.80 (12.02–13.73) | 12.74 (12.00–13.78) | 13.06 (12.13–13.88) | 12.41 (11.44–13.44)* |
| FT3 (pmol/L) | 4.56 (4.19–4.87) | 4.65 (4.37–5.00) | 4.46 (4.10–4.81) | 4.89 (4.64–5.18)** | 4.47 (4.14–4.85) | 4.33 (4.04–4.68)** | 4.50 (4.15–4.82) | 4.80 (4.44–5.03)** | 4.48 (4.11–4.81) | 4.49 (4.16–4.79) | 4.60 (4.19–4.94) | 4.56 (4.29–4.84) |

**Notes:**
BMI, Body Mass Index; FBG, fasting blood glucose; TG, triglycerides; TCH, total cholesterol; HDL-C, high-density lipoprotein cholesterol; LDL-C, low-density lipoprotein cholesterol; TSH, thyroid stimulating hormone; TT4, total thyroid hormone; TT3, total triiodothyronine; FT4, free thyroid hormone; FT3, free triiodothyronine.
* $P < 0.05$.
** $P < 0.001$.

**Table 3 Univariable and multivariable analysis for factors associated with PTC in male and female group, respectively.**

| Variables | Female | | | | Male | | | |
| --- | --- | --- | --- | --- | --- | --- | --- | --- |
| | Univariable | | Multivariable | | Univariable | | Multivariable | |
| | OR | *P* value | OR | *P* value | OR | *P* value | OR | *P* value |
| Age | 1.004 (0.995–1.013) | 0.358 | | | 0.996 (0.984–1.008) | 0.525 | | |
| BMI | 1.117 (1.075–1.160) | <0.001 | 1.045 (1.002–1.089) | 0.038 | 1.09 (1.038–1.144) | 0.001 | 1.005 (0.953–1.061) | 0.848 |
| FBG | 2.57 (2.025–3.262) | <0.001 | 2.543 (1.968–3.286) | <0.001 | 2.059 (1.540–2.754) | <0.001 | 2.136 (1.551–2.941) | <0.001 |
| TG | 1.801 (1.490–2.176) | <0.001 | 1.267 (1.025–1.566) | 0.019 | 1.689 (1.392–2.048) | <0.001 | 1.264 (1.039–1.615) | 0.05 |
| TCH | 0.932 (0.832–1.056) | 0.27 | | | 0.786 (0.660–0.935) | 0.007 | 0.778 (0.626–0.968) | 0.024 |
| HDL-C | 0.305 (0.223–0.418) | <0.001 | 0.422 (0.289–0.616) | <0.001 | 0.096 (0.055–0.169) | <0.001 | 0.154 (0.077–0.308) | <0.001 |
| LDL-C | 1.144 (0.982–1.333) | 0.085 | | | 1.056 (0.856–1.304) | 0.611 | | |

## Serum lipid levels were related to the pathological indicators of PTC patients

Stepwise logistic regression analysis was performed to investigate potential serum lipid biomarkers for tumor size, LNM, or nodule number. The original equation contained eight variables: sex, age, BMI, FBG, TG, TCH, HDL-C, and LDL-C. According to the current findings, TG was positively correlated with the risk for tumor size (OR and 95% CI 1.202 [1.058–1.365], $P = 0.005$). Although male (OR and 95% CI 0.652 [0.507–0.84], $P = 0.001$), HDL-C (OR and 95% CI 0.606 [0.42–0.873], $P = 0.036$), and LDL-C (OR and 95% CI 0.856 [0.734–0.998], $P = 0.047$) were negatively correlated with multifocal, BMI and age were positively linked to the multifocal risk. In addition, male (OR and 95% CI 0.54 [0.43–0.677], $P < 0.001$) and age (OR and 95% CI 0.969 [0.96–0.977], $P < 0.001$) were negatively associated with LNM. The results are displayed in Table 4. Overall, there was a substantial correlation between the pathological signs of PTC patients and blood lipid levels.

## DISCUSSION

Over the previous few decades, despite the concomitant rise in obesity and TC incidence, few studies have focused on the connection between lipid levels and PTC (*Pappa & Alevizaki, 2014*). Here, for the first time, a cross-sectional study was conducted. Data demonstrated that elevated TG and declined HDL-C were linked to increased PTC risk among Chinese of both sexes. Multivariate analysis also demonstrated that FBG, TG, and HDL-C were risk factors for PTC both in males and females.

TG is one of the significant components of body fat. According to recent studies, TG is a potential risk factor for prostate cancer (*Zhu, Hu & Fan, 2022*) and ovarian cancer (*Trabert et al., 2021*). The results of this study suggested that patients with PTC had significantly higher TG levels than controls ($P < 0.05$) in each age and gender group. In addition, TG was shown as a risk factor for PTC both in male and female groups. Meanwhile, higher levels of TG in PTC patients indicated a deficiency of immunity and tumor proliferation (*Zewinger et al., 2020*; *Chen et al., 2020*). Previous experimental studies using *in vivo* and *in vitro* models revealed that TG might cause prostate cancer by altering signaling pathways

**Table 4 Stepwise logistic regression analysis was performed to explore serum lipid biomarkers for tumor size or LNM or nodule number.**

| Variables | | Sig. | Exp (B) | 95% CI for Exp (B) | |
|---|---|---|---|---|---|
| | | | | LOWER | UPPER |
| Tumor size | TG | 0.005 | 1.202 | 1.058 | 1.365 |
| Nodule number | Gender | 0.001 | 0.652 | 0.507 | 0.840 |
| | BMI | 0.015 | 1.042 | 1.006 | 1.078 |
| | Age | <0.001 | 1.021 | 1.013 | 1.030 |
| | HDL | 0.036 | 0.606 | 0.420 | 0.873 |
| | LDL | 0.047 | 0.856 | 0.734 | 0.998 |
| Lymph node metastasis | Gender | <0.001 | 0.540 | 0.430 | 0.677 |
| | Age | <0.001 | 0.969 | 0.960 | 0.977 |

that support carcinogenic processes, including cell growth and proliferation, oxidative stress, inflammation, and cell migration (*Arthur et al., 2016*; *Yue et al., 2014*). *Li et al. (2021)* investigated the effects of palmitic acid stimulation on thyroid cell function using *in vitro* tests. Thyroglobulin, sodium iodide transporter, and thyroid peroxidase have lower mRNA and protein levels when palmitic acid, the most prevalent form of palmitic acid, is stimulated. This might be due to the damage to the synthesis of thyroid hormones (*Li et al., 2021*). However, the underlying mechanisms of how TG mediated progress in PTC still need further illustration.

By encouraging cell migration, proliferation, and invasion, TCH contributes significantly to cancer development (*Nazih & Bard, 2020*). The TCH level was significantly lower in female PTC patients of the youth group ($P < 0.05$). Another retrospective study of TC patients supported this data and showed lower serum cholesterol levels in TC patients, especially in PTC and FTC patients (*Li et al., 2019*). Lower TCH in malignancies may be due to the increased need for cholesterol by tumor cells, which is similar to what has been observed in the acute phase responses of various acute and chronic diseases (*Cabana, Siegel & Sabesin, 1989*). Lipid dysregulation in cancer may be a response in the acute phase brought on by the transmission of cytokines by inflammation-related cells surrounding tumor cells or by the tumor cells themselves (*Blackman, Cabana & Mazzone, 1993*). The sex-specific association between TCH and PTC risk (significant in males but not females) may reflect hormonal or metabolic differences between sexes. Estrogen's role in modulating lipid metabolism and cancer pathways could attenuate the TCH-PTC relationship in females, whereas androgen-related cholesterol utilization in males might enhance this association (*Yue et al., 2014*; *Mendelsohn & Karas, 2005*). Further mechanistic studies are warranted to validate this hypothesis.

Interestingly, the results of this study showed that HDL-C could operate as a preventative measure for PTC. In contrast with the controls, patients with PTC had declined HDL-C levels ($P < 0.05$) in all age and gender groups. HDL-C was shown as an independent PTC biomarker in both male and female groups. A recent study showed that MHR (monocyte/HDL-C) was higher in subjects with PTC, which is an independent PTC risk factor (*Xu*

*et al., 2021*). According to a German primary care provider database with over 60,000 additional patients, lower HDL cholesterol levels positively correlate with cancer (*Loosen et al., 2022*). Sterols and lipids have an excellent affinity for cancer cells, and lipid metabolism has been identified to be essential for cancer signaling (*Cruz et al., 2013*; *Gorin, Gabitova & Astsaturov, 2012*). It was hypothesized that HDL has immunomodulatory, anti-oxidative, anti-apoptotic, and anti-inflammatory properties that may impact the proliferative and inflammatory pathways involved in cancer development (*Onwuka et al., 2020*). However, two recent studies revealed that subjects with metabolic disease had a greater risk of TC when their HDL-C levels were lower (*Kim et al., 2022*; *Nguyen, Kim & Kim, 2022*). Decreased HDL-C is often accompanied by insulin resistance and diabetes (*Rashid, Uffelman & Lewis, 2002*). Although insulin resistance has been identified as a potential contributing factor, the exact mechanism behind the link between HDL-C and TC is yet unknown.

LDL-C is a complex particle made up of a variety of proteins and lipids. The LDL receptor is crucial in endocrine-related tumor cells by improving circulating LDL-C uptake and controlling tumorigenic signaling (*Revilla et al., 2021*). Regarding serum cholesterol, only restricted studies have focused on the relationship between LDL-C and TC risk. A retrospective study revealed lower LDL-C levels in a large cohort of female TC patients and women with metastases in the PTC group (*Li et al., 2019*). However, this study found no noticeable difference in LDL-C between PTC patients and controls in all age groups.

Glucose is the primary source of energy for cells (*Shaw, 2006*). Numerous studies have linked hyperglycemia to an enhanced risk of developing cancer (*Samuel et al., 2018*; *Satija et al., 2015*). According to current knowledge, hyperglycemia has a role in the development and spread of tumors *via* transcription regulators, kinases, growth factors, proteases, oxidoreductases, receptors, developmental proteins, cytokines, and other molecules (*Cai et al., 2022*). In this study, in contrast with the control group, patients with PTC held higher FBG levels ($P < 0.05$), and an increased OR of PTC was associated with FBG both in the univariate and multivariate analysis ($P < 0.001$). It has been reported that hyperglycemia induces an increase in intranuclear nuclear factor (NF)-κB, whose ability to regulate proliferative and anti-apoptotic signaling pathways in thyroid neoplastic cells has been found to play a significant role in TC (*Alkurt et al., 2022*).

The high number of participants is the strength of this study. Retrospective big data analysis revealed the correlation between serum lipid and PTC, providing evidence for clinical diagnosis and healthcare. The current study still had several restrictions. First, the study's single-center retrospective case-control design made it unable to demonstrate whether the link between lipid and PTC is causal or time-dependent. Second, only the first serum lipid levels after admission were collected since they were obtained before surgery and had the fewest influencing factors. However, they cannot represent the daily condition of the patients. Third, information on hormonal contraceptive use in female participants was unavailable, which may partially confound the observed sex-specific associations between lipid profiles and PTC risk. Finally, because only Chinese PTC patients were studied, it is difficult to extrapolate these results to other populations.

## CONCLUSION

The study results revealed that elevated TG and declined HDL-C were related to increased PTC risk among Chinese of both sexes, which may contribute to further investigation concerning diagnosing and preventing this most common type of thyroid cancer.

## ABBREVIATIONS

| | |
|---|---|
| TC | Thyroid cancer |
| PTC | papillary thyroid carcinoma |
| FTC | follicular thyroid carcinoma |
| MTC | medullary thyroid carcinoma |
| ATC | anaplastic thyroid carcinoma |
| TI-RADS | thyroid imaging reporting and data system |
| BMI | body mass index |
| FBG | fasting blood glucose |
| TG | triglyceride |
| HDL-C | high-density lipoprotein cholesterol |
| LDL-C | low-density lipoprotein cholesterol |
| TCH | total cholesterol |
| LNM | lymph node metastasis |
| LNR | lymph node ratio |
| TSH | serum thyroid stimulating hormone |
| TT3 | total triiodothyronine |
| TT4 | total thyroid hormone |
| FT3 | free triiodothyronine |
| FT4 | free thyroid hormone |
| SD | standard deviation |
| OR | Odds Ratio |
| CIs | confidence intervals |
| MHR | Monocyte high-density lipoprotein cholesterol ratio |

### Funding

The authors received no funding for this work.

### Competing Interests

The authors declare that they have no competing interests.

### Author Contributions

- Zike Zhang conceived and designed the experiments, analyzed the data, prepared figures and/or tables, and approved the final draft.

- Xingyu Lan performed the experiments, prepared figures and/or tables, and approved the final draft.
- Long You performed the experiments, authored or reviewed drafts of the article, and approved the final draft.
- Dongsheng Han performed the experiments, analyzed the data, prepared figures and/or tables, authored or reviewed drafts of the article, and approved the final draft.
- Hui Tang performed the experiments, analyzed the data, prepared figures and/or tables, authored or reviewed drafts of the article, and approved the final draft.
- Ying Zhao analyzed the data, prepared figures and/or tables, authored or reviewed drafts of the article, and approved the final draft.
- Xiao Hu conceived and designed the experiments, authored or reviewed drafts of the article, and approved the final draft.

### Human Ethics

The following information was supplied relating to ethical approvals (*i.e.*, approving body and any reference numbers):

The Ethics Committee of the First Affiliated Hospital of Medical College at Zhejiang University approved the study (IIT20230089A).

### Data Availability

Raw data is available in the Supplemental Files.

### Supplemental Information

Supplemental information for this article can be found online at http://dx.doi.org/10.7717/peerj.19644#supplemental-information.

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
