# Peer review of "Association between serum lipid and papillary thyroid cancer: a retrospective study in China"

_PeerJ, doi:10.7717/peerj.19644_

## Round 0.1 · original submission · Minor Revisions

**Language Note:** The review process has identified that the English language must be improved. PeerJ can provide language editing services - please contact us at [email protected] for pricing (be sure to provide your manuscript number and title). Alternatively, you should make your own arrangements to improve the language quality and provide details in your response letter. – PeerJ Staff

Reviewer 1 ·

Basic reporting

Sufficient background information is given.

Experimental design

The experimental design is appropriate, one information should be clarified (see additional comments).

Validity of the findings

Findings are valid.

Additional comments

Based on the evaluation of BMI and various blood parameters obtained from patients with papillary thyroid cancer and healthy controls, the study describes that elevated TG and declined HDL-C were related to increased PTC risk. The study is well conducted and methods and data well described.
Did the females take any hormonal contraceptives that will influence their lipid profiles?
What is the explanation that the OR for TCH is present only in males?
L.118: the description “larger patients” is not clear.

Reviewer 2 ·

Basic reporting

Several sentences are excessively long, which may impede readability. I recommend using an AI-based editing tool or a professional language improvement service to enhance clarity and flow.
No further comments

Experimental design

No further comment

Validity of the findings

No turther comment

Additional comments

The current study investigated the relationship between serum lipids and papillary thyroid cancer. Overall, the manuscript is well-designed and holds significant clinical relevance. Here are my comments:
Major Comments:
1. Language. Several sentences are excessively long, which may impede readability. I recommend using an AI-based editing tool or a professional language improvement service to enhance clarity and flow.
Minor Comments:
1. Introduction. Revise the sentence: “Results showed that elevated TG and declined HDL-C were related to increased PTC risk among Chinese of both sexes, which may contribute to further investigation concerning diagnosing and preventing this most common type of thyroid cancer.” to: “The study could contribute to further investigation into diagnosing and preventing papillary thyroid cancer, the most common type of thyroid cancer.”
2. Methods. Change the sentence: “we allocated an alias to data immediately after export” to: “Anonymization was conducted prior to further analysis.”, assuming this accurately reflects the intended meaning.
3. Methods. Please ensure that “chi” in “chi-square test” and “t” in “t test” are italicized in accordance with standard scientific notation.
4. Figure. Clarify whether the lines in Figure 1 represent 95% confidence intervals (95% CI), standard error of the mean (SEM), or standard deviation (SD).

---

## Round 0.2 · accepted · Accept

Thank you for your efforts to address the reviewer comments. Your manuscript has been reviewed and is now suitable for publication.

Reviewer 1 ·

Basic reporting

No comments.

Experimental design

Comments have been addressed.

Validity of the findings

Comments have been addressed.

Additional comments

I have no further comments.

Reviewer 2 ·

Basic reporting

no further comments

Experimental design

no further comments

Validity of the findings

no further comments

Additional comments

no further comments